# Latitudinal Variation in the Pattern of Temperature-Dependent Sex Determination in the Japanese Gecko, *Gekko japonicus*

**DOI:** 10.3390/ani12080942

**Published:** 2022-04-07

**Authors:** Shuran Li, Zhiwang Xu, Laigao Luo, Jun Ping, Huabin Zhou, Lei Xie, Yongpu Zhang

**Affiliations:** 1College of Life and Environmental Science, Wenzhou University, Wenzhou 325035, China; lishuran@wzu.edu.cn (S.L.); 20451334037@stu.wzu.edu.cn (Z.X.); pingjun1@cib.ac.cn (J.P.); zhb@wzu.edu.cn (H.Z.); xl@wzu.edu.cn (L.X.); 2School of Biological Science and Food Engineering, Chuzhou University, Chuzhou 239000, China; llaigao@chzu.edu.cn

**Keywords:** geographic variation, reptile, sex determination, sex ratio, thermal environment

## Abstract

**Simple Summary:**

In egg-laying lizards, sex is determined by genetic factors in species with sex chromosomes or egg incubation temperatures in species without sex chromosomes, i.e., temperature-dependent sex determination (TSD). Surprisingly, recent studies find sex chromosomes and TSD co-occur in the same species in some lizards. The Japanese gecko from Japan may be this case. However, Japanese gecko with TSD from a Chinese population does not have sex chromosomes, suggesting that the pattern of TSD in this gecko may vary among populations. We incubated gecko eggs from three populations in China at constant temperatures of 24, 26, 28, 30, and 32 °C to quantify the sex determination pattern. We found that the temperature yielding an equal number of sons and daughters of the low-latitude population was lower than that of the two high-latitude populations. Moreover, the low-latitude population had a narrower temperature range producing mixed sex offspring at lower temperatures, but a wider range at higher temperatures. Sex ratio was almost 1:1 for the low-latitude population when incubated from 26 to 30 °C. Conversely, more male offspring were produced at 28 or 30 °C in the two high-latitude populations. Our study may provide an interesting system to explore the evolution of sex determination mechanisms in animals.

**Abstract:**

Identifying latitudinal variation in the pattern of temperature-dependent sex determination (TSD) may provide insight into the evolution of sex determining system in vertebrates, but such studies remain limited. Here, we quantified TSD patterns of three geographically separated populations of the Japanese gecko (*Gekko japonicus*) along the latitudinal cline of China. We incubated gecko eggs from the three populations at constant temperatures of 24, 26, 28, 30, and 32 °C to quantify the TSD pattern. Our study demonstrated that *G. japonicus* exhibited a FMF pattern of TSD, with the low and high incubation temperatures yielding significantly female-biased hatchlings, and the medium temperatures producing male-biased hatchlings. More interestingly, we found latitudinal variations in the TSD pattern in terms of pivotal temperatures (T_piv_s), transitional range of temperatures (TRT), and the sex ratios at the medium temperatures. The T_piv_s for the low-latitude population were lower than those for the two high-latitude populations. The low-latitude population has a narrower FM TRT, but a wider MF TRT. The sex ratio is almost 50:50 for the low-latitude population when eggs were incubated from 26 to 30 °C. Conversely, the sex ratio is male-biased for the two high-latitude populations at 28 or 30 °C. Therefore, *G. japonicus* may provide an interesting system to explore the evolution of TSD in reptiles given the diversity of TSD patterns among populations.

## 1. Introduction

Sex determination is a basic question that has attracted a great deal of interest from biologists [1,2,3]. Vertebrates exhibit a diverse range of sex-determining mechanisms from environmental sex determination (ESD) to genotypic sex determination (GSD) [3,4]. Temperature-dependent sex determination (TSD), a typical type of ESD, is particularly common in reptiles, involving all tuataras and crocodilians, most turtles, and some lizards [3]. Identifying the pattern of sex determination is essential to understanding the evolution of TSD in vertebrates.

In reptiles with TSD, three typical patterns have been described according to the reaction norms of offspring sex ratio versus temperatures during embryonic development: (1) male–female (MF or TSD Ia), with a prevalence of males produced at low incubation temperatures and females produced at high temperatures; (2) female–male (FM or TSD Ib), which is opposite to the FM pattern; and (3) female–male–female (FMF or TSD II), indicating two female-skewed sex ratios are produced at both high and low temperatures, while a higher percentage of males is produced at intermediate temperatures [3]. Two key parameters have been introduced to characterize the mathematical models of the TSD patterns. The best known parameter is the pivotal temperature (T_piv_), at which hatchling sex ratios are 1:1 ratios [5]. Another important parameter of TSD reaction norm is the transitional range of temperature (TRT), which refers to the range of incubation temperatures producing mixed sex and has received increasing attention [6].

The patterns of TSD not only differ among species, but also within species, such as latitudinal variation in the TSD reaction norms. For example, some research has found T_piv_ is correlated with latitudes, with higher T_piv_s at higher latitudes at some freshwater turtles [5,7]. However, the opposite pattern has also been observed, as the T_piv_ is higher in the low-latitude population than the high-latitude population in a freshwater turtle (*Mauremys mutica*) [8,9], or does not show geographic variation in sea turtles of *Caretta caretta* and *Chelonia mydas* [10,11]. Recent studies also find inconsistent geographic variation patterns in TRT of some sea turtles, as the TRT is broader in low-latitude papulations than in high latitude papulations *C. caretta* [6] and *C. mydas* [12], while a wider TRT has been detected in both low- and high-latitude papulations in *Natator depressus* [12]. Therefore, our understanding of latitudinal variation of TSD patterns remains limited and merits further study.

The Japanese gecko (*Gekko japonicus*) is a TSD gekkonid lizard [13]. The pattern of temperature-dependent sex determination slightly varies among populations, with female-biased hatchlings at low (24 °C) and high (32 °C) incubation temperatures, male-biased hatchlings at medium incubation temperatures (28 °C) in a Japanese population [13], and even a sex ratio of hatchlings at medium incubation temperatures (28 °C) in a Chinese population from Nanjing [14]. In addition, heteromorphic sex chromosomes and TSD co-occur to the Japanese population [13], but geckos from a Chinese population of Nanjing do not have heteromorphic sex chromosomes [15]. These between-population differences in karyotype indicate that the TSD pattern in this species may be even more complicated than we think. Given that the Japanese gecko is a widespread species experiencing distinct climates in eastern Asia, it is interesting to consider whether the TSD pattern differs among populations or not on a broad scale. The answers to this question may improve our understanding around how the populations adapt to local thermal environments and form the basis for further studies on the proximate and ultimate causes of TSD in reptiles. Therefore, this species may provide a unique system to investigate such questions. Here, we quantified the TSD pattern in three geographically separated populations of the Japanese gecko along the latitudinal cline: the Yancheng population (33°43′ N, 120°23′ E), the Chuzhou population (32°17′ N, 118°17′ E), and the Wenzhou population (27°45′ N, 120°36′ E) of China. To quantify the TSD pattern, we incubated gecko eggs from the three populations at constant temperatures of 24, 26, 28, 30, and 32 °C, and identified the sex of hatchlings. More specifically, we aim to test for the hypothesis that the T_piv_ of sex determination is higher for high-latitude populations than low-latitude populations, but TRT is wider in the low-latitude population.

## 2. Materials and Methods

### 2.1. Study Species and Animal Collection

The Japanese gecko (*Gekko japonicus*) is a small nocturnal lizard (adult snout-vent length < 80 mm) mainly distributed in eastern China, southern Korea, and Japan [16]. Females usually lay one or two clutches from May to August, with typically two rigid-shelled eggs in each clutch [17,18]. In *G. japonicus*, a southerly population had larger body size and produced larger eggs than two northerly populations [19]. 

We collected gravid female geckos in early May 2016 from three different populations in China: a northern population (Yancheng, *n* = 181), a central population (Chuzhou, *n* = 80), and a southern population (Wenzhou, *n* = 210). Air temperatures were extracted from data of the nearest climate station (<8 km) for each location [Dataset of daily surface observation values in individual years (1981–2010) in China, China Meteorological Data Service Center, http://data.cma.cn/, accessed on 14 July 2018]. The dataset provided daily mean air temperatures, daily maximum air temperatures, and daily minimum air temperatures of 30-year means. 

At each location, ten small button-sized temperature data loggers (iButtons, DS1921G, MAXIM Integrated Products, Ltd., San Jose, CA, USA) were programed to record habitat (potential nest) temperatures every three hours from early May to late July automatically. These iButtons were randomly placed in crevices of the buildings where gravid lizards were found as it was impossible to search nests of *G. japonicus* in private buildings. Because some buildings were renovated or even destroyed, we only retrieved five iButtons from Yancheng, none from Chuzhou, and seven from Wenzhou.

### 2.2. Egg Incubation and Hatchling Husbandry

Lizards were transported to the laboratory in Wenzhou University. Each lizard was kept individually in a plastic box (30 cm × 20 cm × 15 cm) which was lined with plastic wrap, allowing the eggs to be easily removed, and contained a piece of cardboard for shelter. They were placed in a room where temperature was kept around 25 °C and light was set 0630 on and 1830 off. A heat source was provided underneath the boxes between 1600 and 2400 for thermoregulation. Food (larvae of *Tenebrio molitor* and *Acheta domesticus* dusted with vitamins and minerals) and water were provided *ad libitum*. 

Each box was checked four times daily for freshly laid eggs. As the two eggs in each clutch usually adhered together, they were weighted together (ML204, Mettler Toledo, Zurich, Switzerland, ±0.001 g), and each egg mass was calculated by multiplying the total mass by the percentage of egg volume. Egg volume was estimated following the equation for the volume of ellipsoid: V = 4/3π × length/2 × (width/2)^2^. The length and width of each egg were measured with an Absolute Digimatic Caliper (Mitutoyo, Kawasaki, Japan, ±0.01 mm). All females were released to the location of capture in August.

The two eggs in each clutch were incubated together in a plastic box (10 cm ×10 cm × 3 cm) and half buried in moist vermiculite (−220 kPa). Each box was covered by a lid with four holes (2 mm diameter), and then assigned to one of five incubators (KB400, Binder GmbH, Tuttlingen, Germany) with constant temperatures (24, 26, 28, 30, and 32 °C). Lost water was added to the vermiculite every week. The incubators were checked for new hatchlings four times daily when the incubation period lasted around 90% at each temperature. Once hatched, each hatchling was painted a unique number on back with marker pen for identification. Hatchlings were housed in cloth cylinder cages (40 cm in diameter, 50 cm in height), lined with a piece of cardboard for shelter, and kept following the husbandry procedure for females as described above. Each cage contained no more than 15 hatchlings from the three populations and checked four times every day. The faded number of the hatchlings was re-painted. Dead hatchlings were frozen in −80 °C refrigerator (CryoCube F570, Eppendorf AG, Barkhausenweg, Hamburg, Germany) immediately once found. 

### 2.3. Sex Identification

Hatchlings dying in one month were sexed through histological analysis of gonads following Tokunaga [13]. Gonads were excised from dead hatchlings immediately once found and fixed in Bouin solution. After dehydrated with a serial change in alcohol, the specimens were embedded in paraffin and sliced to 5 μm-thick sections. The slices were stained with hematoxylin–eosin and observed under stereomicroscope (Nikon SMZ800N, Tokyo, Japan). Sexes of hatchings dying in one to two months were determined by observing the shape of gonads stained with hematoxylin–eosin under a stereomicroscope. Hatchings that died in two to four months were sexed in a similar way except staining. Living hatchlings were sexed by everting the hemipenes in males after four months [14] (Appendix A). 

### 2.4. Statistical Analyses

One-way ANOVA was used to compare differences of habitat temperatures (mean daily mean, mean daily maximum, mean daily minimum, and mean daily range) between populations. To investigate the effects of incubation temperature, population, and their interaction on hatching success, we used a generalized linear mixed model following a binomial distribution and a logit link function with maternal identity as a random factor. Chi square goodness-of-fit tests were used to evaluate whether the sex ratio of hatchlings from each treatment departed from 1:1. A binomial logistic regression was used to analyze the relationship between egg mass and hatchling sex. All analyses above were performed in IBM SPSS 20. 

Meanwhile, the “tsd” function in R package “embryogrowth” (v 8.2) [20] was used to estimate the thermal reaction norm for sex ratio of each population. Logistic and flexit models were used to search for the best fit of the curve according to Abreu-Grobois et al. [21]. The best fit model was selected with corrected Akaike information criterion (AICc) and Akaike weight. The Bayesian Markov chain Monte Carlo (MCMC) procedure was used to estimate the pivotal temperatures and the transitional range of temperatures (TRT), defined as the range of incubation temperatures producing sex ratios from 5% to 95% [6]. 

## 3. Results

### 3.1. Air and Habitat Temperatures

Air temperatures increased with decreasing latitude, although Chuzhou and Wenzhou had similar mean air temperatures from April to July (Appendix A). The habitat temperatures of Yancheng were significantly lower, but fluctuated more dramatically than those of Wenzhou (Table 1). 

### 3.2. Hatchling Sex

Hatching success was high (86.7–98.4%) across all treatments, and was not significantly affected by incubation temperatures, populations and their interaction (incubation temperature: *F*_4,830_ = 0.06, *p* = 0.993; population: *F*_2,830_ = 0.21, *p* = 0.814; interaction: *F*_8,830_ = 1.24, *p* = 0.271).

The sex ratio was not significantly related to egg mass (Wald test, χ^2^ = 0.472, *p* = 0.492). We found significantly female-biased sex ratios in hatchlings incubated at 24, 26, and 32 °C for the Yancheng and Chuzhou populations, but at 24 and 32 °C for the Wenzhou population (Figure 1, Table 2). The sex ratio was significantly male-biased (but not 100%) in hatchlings incubated at 28 °C for the Yancheng population, and at 30 °C for the Chuzhou population (Figure 1, Table 2). In contrast, hatchling sex ratio was almost 50:50 at incubation temperatures from 26 to 30 °C in the Wenzhou population (Figure 1, Table 2).

Logistic model best fit the data of each population based on AICc selection (Appendix A). The FM and MF T_piv_s were higher in the Yancheng (FM T_piv_ = 26.5 °C, MF T_piv_ = 30.7 °C, Figure 1, Table 3) and Chuzhou (FM T_piv_ = 27.2 °C, MF T_piv_ = 30.8 °C, Figure 1, Table 3) populations than those in the Wenzhou population (FM T_piv_ = 25.1 °C, MF T_piv_ = 29.3 °C, Figure 1, Table 3). The FM TRT was slightly wider in Yancheng and Chuzhou populations, while the MF TRT was largely wider in the Wenzhou population (Figure 1, Table 3).

## 4. Discussion

Our study demonstrated that the low (24 °C) and high (32 °C) incubation temperatures yielded significantly female-biased hatchlings, while the medium temperatures (around 28 °C) produced male-biased hatchlings or an even sex ratio of hatchlings in *G. japonicus*. This pattern is generally consistent with the findings from previous studies on different populations of this species [13,14]. Nonetheless, we found latitudinal variation in the TSD pattern of this species in terms of T_piv_s, TRT, and the sex ratios at the medium temperatures.

The MF T_piv_ for the low-latitude (Wenzhou) population is lower than that for the two high-latitude (Yancheng and Chuzhou) populations. This finding is contradicted with the hypothesis that the T_piv_ of sex determination is higher for high-latitude populations than low-latitude populations. A similar latitudinal variation in the TSD pattern has been reported in some turtles [5,7], but not in others [8,9]. Despite the between-species discrepancy in latitudinal pattern of TSD, the underpinning notion to explain this geographic variation is the same: the T_piv_ is correlated with the thermal environments where the species have adapted [22]. For those species at lower latitudes with higher nest temperatures, natural selection favor higher T_piv_s, consistent with the expectation from the Fisherian theory of sex ratio evolution [5]. Alternatively, those species at higher latitudes have higher T_piv_s, perhaps because high-latitude females nest in exposed locations with higher rather than lower temperatures at high latitudes [5,23]. Unfortunately, we do not have data on nest temperatures on this species due to the difficulty to locate nests in the field, which therefore precludes us from testing whether female geckos at a higher latitude select nest sites with higher temperatures. Moreover, unlike most turtles with a MF pattern of TSD, and therefore only a MF T_piv_, *G. japonicus* show a FMF pattern of TSD, and therefore FM and MF T_piv_s. This phenomenon makes the response of offspring sex to environmental temperatures more complicated, because both FM and MF T_piv_s are located within the range of habitat (potential nest) temperatures, and thus related to offspring sex ratio in the field. Our results show that the FM T_piv_ is lower at low-latitude populations than high-latitude populations. How this between-population difference in the FM T_piv_ contributes to sex ratios in the field remains unknown. One possible explanation is that female offspring produced in the early nesting season benefit more than males from a longer growing season as they attain sexually maturity with larger size early in their second summer of life, thus enhancing their reproductive success [17], while contemporaneous males may be unable to compete successfully with the much larger conspecifics [24]. Such selection is likely weakened in the low-latitude population with quite a long growing season and relatively small seasonal variation. This question deserves further study to unravel the importance of the FM T_piv_ in determining the sex ratio of offspring in TSD species. In addition, previous studies demonstrate that latitudinal patterns of dual T_piv_s vary among TSD species. The two pivotal temperatures for high-latitude populations are both higher and lower than those for low-latitude populations [5], but the dual pivotal temperatures are not related with latitudes in the Australian water dragon (*Physignathus lesueurii*) [25].

Although TRT has received less attention than T_piv_s, a recent study indicates that widening TRT can decrease selection on T_piv_s, and that TRT may reflect correlations of adaptation in TSD with local temperature better than T_piv_s [6]. As a FMF TSD species, *G. japonicus* has two TRT encompassing the T_piv_s, and shows a complex variation in TRT. Compared to the two high-latitude populations, the low-latitude population has a narrower FM TRT, but a wider MF TRT (Table 3, Figure 1). The proportion of mixed sex nests would be greater in populations with wider TRT, which may reduce the risk of sex ratio bias and increase population viability [26]. Therefore, the reverse pattern of the two TRT in *G. japonicus* may correspond to the adaptation to local climatic conditions of different populations, as potential nest temperatures in the high-latitude population (Yancheng) likely fall within the FM TRT, while that in the low-latitude population is around its MF TRT, which is consistent with the findings in sea turtles *C. caretta* [6] and *C. myda* [12] with a MF pattern. Nevertheless, an interspecies comparison study does not discover the apparent variation in TRT (0.9–1.8 °C) among nine crocodilians with a FMF pattern [27]. Overall, it is urgent to take TRT into account due to insufficient data on geographic variation of TRT in FMF species and the importance of TRT to understanding the adaptive nature of TSD [5,6,25].

Another significant between-population difference in TSD patterns is the sex ratio at medium temperatures around 28 °C. The sex ratio is almost 50:50 for the low-latitude population when eggs were incubated at temperatures from 26 to 30 °C. In contrast, the sex ratio is male-biased for the two high-latitude populations at temperatures of 28 or 30 °C (Figure 1). The answers to this between-population difference are unknown, but may be related to the following issues. Firstly, given that some populations possess heteromorphic sex chromosomes, but other populations do not in *G. japonicus*, as shown by previous studies [13,15], this between-population difference could be related to the occurrence of sex chromosomes in specific populations. To solve this issue, it would be of great interest to identify the karyotype and its relationship with sex ratios in each population. Secondly, this between-population difference could be due to the difference in maternally derived yolk steroid hormones, because maternally derived yolk steroid hormones can influence offspring sex in a number of TSD reptiles [28,29,30,31]. In our species, maternally derived yolk steroid hormones interact with incubation temperatures to affect offspring sex ratio, with substantial effects of yolk hormones on offspring sex ratio at the medium temperature of 28 °C [14]. Thirdly, this between-population difference could be related to thermal regimes at these populations. Although we do not have data on nest temperatures of this species, the habitat temperatures of these populations provide some cues. For example, habitat temperatures could fluctuate more widely in high-latitude populations than low-latitude populations (Table 1). Spatiotemporal variation in the strength of sex-specific selection on TSD may induce evolutionary divergence in sex determination between populations or species [32,33]. A previous study on the Australian snow skink (*Niveoscincus ocellatus*) demonstrates that TSD is favored in lowland populations, while GSD is favored in highland populations with a relatively cold and more variable climate [34]. Unlike the snow skink, low-latitude populations of *G. japonicus* with a relatively stable climate show a TSD pattern with temperatures ranging from 26 to 30 °C, producing a 50:50 sex ratio, which seems to be closer to the GSD pattern. Despite increasing evidence that demonstrated temperature fluctuations experienced by embryos can have significant effects on offspring sex ratios [35,36,37], our understanding of the role of temperature fluctuations in the TSD evolution is elusive. This is because there is a lack of clarity around the relationship between temperature fluctuations and offspring sex, probably due to the unpredictable daily and seasonal variations in nest temperatures [29,38].

## 5. Conclusions

More generally, our study identifies a FMF pattern of TSD that differs among populations in an Asian gecko. This may provide an interesting system to explore the evolution of TSD and GSD in reptiles, but many open questions remain to be answered. Future studies should focus on detailed information on maternal nest selection, nest temperatures, the relationship between nest temperatures and sex ratio, and the genetic basis of TSD patterns among these populations. Answers to these questions help us fully understand the ecological and evolutionary significances of this FMF pattern of TSD, therefore highlighting the evolution of TSD and GSD in reptiles.

## Figures and Tables

**Figure 1 animals-12-00942-f001:**
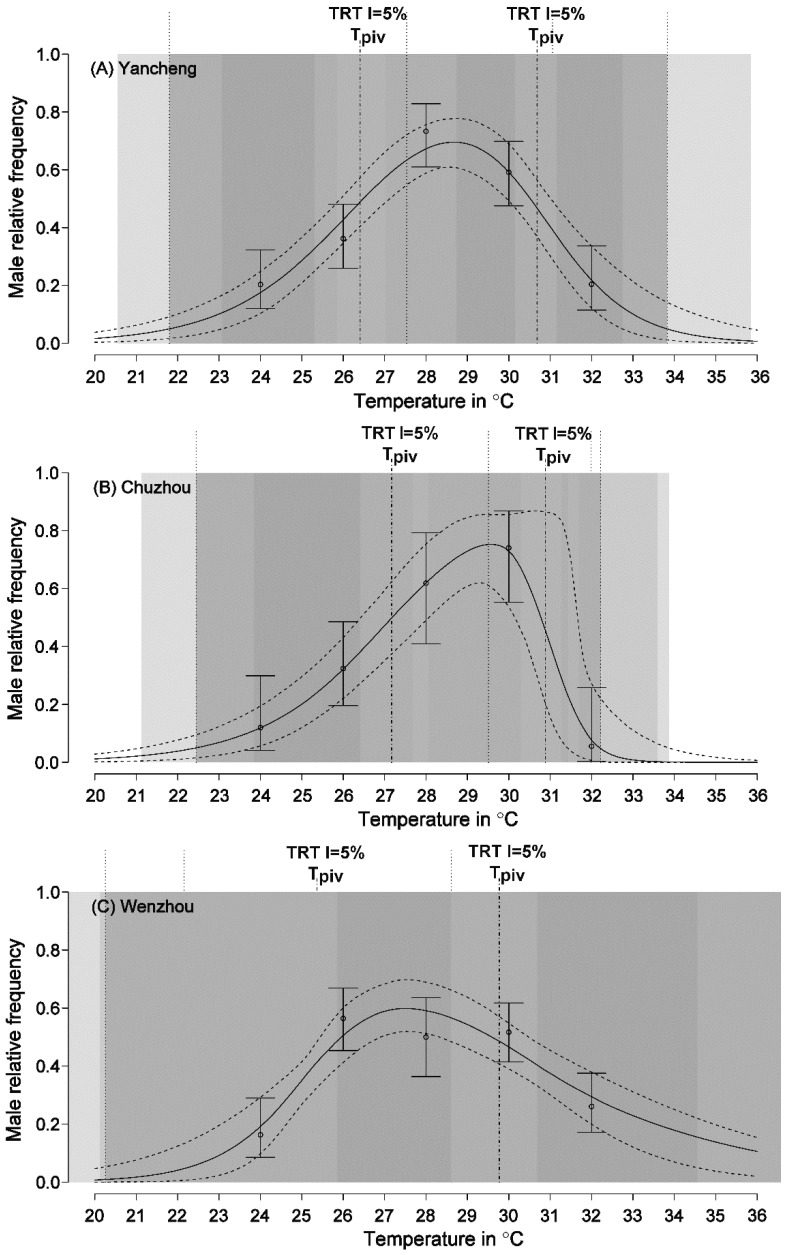
Patterns of temperature-dependent sex determination in *Gekko japonicus* from three populations modeled by a logistic function and fitted using Bayesian MCMC. (**A**) Yancheng, (**B**) Chuzhou, and (**C**) Wenzhou. The points indicate observed sex ratios and the bars represent their 95% confidence intervals. The vertical dash-dotted lines depict pivotal temperatures (T_piv_). The dark gray zones represent the transitional range of temperatures (TRT) and the light gray zone represents the 95% confidence of the TRT. The vertical dotted lines represent the lower and higher limits of TRT. The plain solid curves depict the thermal reaction norm for sex ratios associated with dashed curves which represent a 95% confidence interval.

**Table 1 animals-12-00942-t001:** Habitat (potential nest) temperatures of *Gekko japonicus* from Yancheng (northern population) and Wenzhou (southern population). Habitat temperatures of each location were recorded every three hours from early May to late July by ten iButtons (DS1921G, MAXIM Integrated Products, Ltd., San Jose, CA, USA) which were randomly placed in crevices of the buildings where gravid lizards were found. Data are expressed as means ± SE. *p* values in bold indicate significant differences.

Temperatures (°C)	Yancheng	Wenzhou	Statistic Value
maximum	29.4 ± 0.6	30.8 ± 0.3	*F*_1, 10_ = 4.462, *p* = 0.061
mean	25.0 ± 0.3	27.8 ± 0.2	*F*_1, 10_ = 52.504, ***p* < 0.001**
minimum	22.0 ± 0.1	25.5 ± 0.3	*F*_1, 10_ = 71.414, ***p* < 0.001**
range	7.4 ± 0.5	5.3 ± 0.5	*F*_1, 10_ = 8.440, ***p* = 0.016**

**Table 2 animals-12-00942-t002:** Sex ratio of *Gekko japonicus* from different populations under each incubation temperature (Yancheng, northern population; Chuzhou, central population; Wenzhou, southern population). *df* indicates degree of freedom. *p* values in bold indicate significant differences.

Population	Incubation Temperature/°C	Female: Male (Unidentified)	Sex Ratio Departure Test
Yancheng	24	47: 12 (0)	*χ*^2^ = 20.763, *df* = 1, ***p* < 0.001**
	26	44: 25 (2)	*χ*^2^ = 5.232, *df* = 1, ***p* = 0.022**
	28	16: 44 (1)	*χ*^2^ = 13.067, *df* = 1, ***p* < 0.001**
	30	29: 42 (2)	*χ*^2^ = 2.380, *df* = 1, *p* = 0.123
	32	39: 10 (0)	*χ*^2^ = 17.163, *df* = 1, ***p* < 0.001**
Chuzhou	24	22: 3 (0)	*χ*^2^ = 14.440, *df* = 1, ***p* < 0.001**
	26	25: 12 (3)	*χ*^2^ = 4.568, *df* = 1, ***p* = 0.033**
	28	8: 13 (0)	*χ*^2^ = 1.190, *df* = 1, *p* = 0.275
	30	7: 20 (2)	*χ*^2^ = 6.259, *df* = 1, ***p* = 0.012**
	32	17: 1 (0)	*χ*^2^ = 14.222, *df* = 1, ***p* < 0.001**
Wenzhou	24	41: 8 (3)	*χ*^2^ = 22.224, *df* = 1, ***p* < 0.001**
	26	34: 44 (6)	*χ*^2^ = 1.282, *df* = 1, *p* = 0.258
	28	24: 24 (1)	*χ*^2^ = 0.000, *df* = 1, *p* = 1.000
	30	43: 46 (0)	*χ*^2^ = 0.101, *df* = 1, *p* = 0.750
	32	51: 18 (1)	*χ*^2^ = 15.783, *df* = 1, ***p* < 0.001**

**Table 3 animals-12-00942-t003:** Quantiles (2.5%, 50%, and 97.5%) for pivotal temperature (T_piv_) and transitional range of temperatures (TRT) for *Gekko japonicus* from different populations using Bayesian MCMC with a logistic model.

Population	Parameters	Female to Male			Male to Female		
		2.5%	50%	97.5%	2.5%	50%	97.5%
Yancheng	T_piv_ (°C)	25.8	26.4	27.0	30.2	30.7	31.2
	TRT(°C)	6.7	9.3	11.6	4.2	6.3	10.0
Chuzhou	T_piv_ (°C)	26.4	27.2	28.1	30.3	30.9	31.8
	TRT(°C)	6.4	9.6	11.7	0.6	2.6	6.3
Wenzhou	T_piv_ (°C)	25.1	25.7	26.4	29.3	30.0	30.7
	TRT(°C)	4.7	8.1	11.5	8.5	10.9	19.4

## Data Availability

The data presented in this study are available in Table 2.

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
