# Peer review of "Latitudinal Variation in the Pattern of Temperature-Dependent Sex Determination in the Japanese Gecko, Gekko japonicus"

_animals, 2022, doi:10.3390/ani12080942_

Round 1
Reviewer 1 Report
The authors collected eggs from the same species across different latitudes to examine if there was variation in temperature dependent sex determination by incubating the eggs at different temperature. They supplemented this experiment with observational data on the average climates of the locations where the eggs were collected to assess if climatic variables correlated with TRT or Tpiv. Their explanation of the project was well done and focused on the driving aspects of the results they saw, though some of the ecological or downstream consequences/impacts could be strengthened for the differences they saw. Doing so would add support to the rest of the paper. Finally, the beginning of the introduction leans heavily on GSD to the point where readers would think that you measured that as well. So I would recommend a softening or limiting of the GSD conversation a little to what is needed for the experiment shown. That information could be useful in the discussion though for the future directions of this type of research.
While they do include the data in table 2, this table is missing the ages of the offspring that were measured. If this could be uploaded as an excel file that would be great. Below are some specific line comments.
44: A diverse range of?
59: Specify that only some species have Tpiv at high temperature. The current reading suggest that correlation is what always happens, while you give examples shortly afterwards that it does not always occur.
75 How are you defining medium incubation temps? Is it relative to local temperatures?
82 Why is it an interesting question beyond them being widespread? What does them being widespread allow you to know/infer?
176: Table 1, is this data from the climate stations or iButtons>
200: Figure 1 If you could make the text larger that would really help with the ability to read it, especially the labels.
239-242 Can you give a suggestion on what the ecological/environmental driver may be of this phenomenon?
Reviewer 2 Report
Comments on the manuscript:
“Latitudinal variation in the pattern of temperature-dependent sex determination in the Japanese gecko, Gekko japonicus”
In oviparous lizards, sex is determined by genetic factors or by temperature. There are species with sex chromosomes whose sex is determined by temperature, such as Gekko japonicus. However, Chinese populations of G. japonicus whose sex is determined by temperature are devoid of sex chromosomes, which suggests that the sex determination of this species is variable depending on the population.
The aim of the study presented here is to verify whether the pivot temperatures of sex determination are higher in high latitude than in low latitude, but the transitional range of temperatures is wider in the low latitude population.
This well-conducted and well-presented work addresses the question of sex determination in reptiles. This determination being carried out genetically (sex chromosomes) or ecophysiologically (temperature effects), or very probably in an intermediate way. Besides the fact that this work provides knowledge on the determination of the sex of reptiles, it seems important at a time when climate change inevitably affects the biology of species. It seems to me that this manuscript could be published after some minor corrections.
Title: use italics to write “Gekko japonicus”
Page 2, lines 77, 78, 79: is it “herteromophic” or “heteromorphic”?
Page 3, line 111: explain what are and what is the use of iButtons (all the readers don’t know these methods)
Page 4, line 147. “…histological analysis of gonads following Tokunaga”: Explain the method.
